# Dietary Acid Load and Its Interaction with IGF1 (rs35767 and rs7136446) and IL6 (rs1800796) Polymorphisms on Metabolic Traits among Postmenopausal Women

**DOI:** 10.3390/nu13072161

**Published:** 2021-06-23

**Authors:** Sook Yee Lim, Yoke Mun Chan, Vasudevan Ramachandran, Zalilah Mohd Shariff, Yit Siew Chin, Manohar Arumugam

**Affiliations:** 1Department of Dietetics, Faculty of Medicine and Health Sciences, Universiti Putra Malaysia (UPM), Serdang 43400, Malaysia; l.sookyee@yahoo.com; 2Research Center of Excellence Nutrition and Non-Communicable Diseases, Faculty of Medicine and Health Sciences, Universiti Putra Malaysia (UPM), Serdang 43400, Malaysia; chinys@upm.edu.my; 3Malaysian Research Institute on Ageing, Universiti Putra Malaysia, Serdang 43400, Malaysia; 4Centre for Research, Bharath Institute of Higher Education and Research, 173, Agaram Main Rd, Selaiyur, Chennai, Tamil Nadu 600073, India; 5Department of Nutrition, Faculty of Medicine and Health Sciences, Universiti Putra Malaysia (UPM), Serdang 43400, Malaysia; zalilahms@upm.edu.my; 6Department of Orthopedics, Faculty of Medicine and Health Sciences, Universiti Putra Malaysia (UPM), Serdang 43400, Malaysia; a_manohar@upm.edu.my

**Keywords:** interleukin 6, insulin growth factor-1, gene polymorphism, acid-base homeostasis, hypertension, diabetes

## Abstract

The objective of this study was to explore the effects of dietary acid load (DAL) and IGF1 and IL6 gene polymorphisms and their potential diet–gene interactions on metabolic traits. A total of 211 community-dwelling postmenopausal women were recruited. DAL was estimated using potential renal acid load (PRAL). Blood was drawn for biochemical parameters and DNA was extracted and Agena^®^ MassARRAY was used for genotyping analysis to identify the signalling of IGF1 (rs35767 and rs7136446) and IL6 (rs1800796) polymorphisms. Interactions between diet and genetic polymorphisms were assessed using regression analysis. The result showed that DAL was positively associated with fasting blood glucose (FBG) (β = 0.147, *p* < 0.05) and there was significant interaction effect between DAL and IL6 with systolic blood pressure (SBP) (β = 0.19, *p* = 0.041). In conclusion, these findings did not support the interaction effects between DAL and IGF1 and IL6 single nucleotide polymorphisms (rs35767, rs7136446, and rs1800796) on metabolic traits, except for SBP. Besides, higher DAL was associated with higher FBG, allowing us to postulate that high DAL is a potential risk factor for diabetes.

## 1. Introduction

Metabolic syndrome (MS) is a cluster of interrelated and heritable metabolic traits, which collectively impart unsurpassed risk for type 2 diabetes mellitus (T2DM) and atherosclerotic cardiovascular disease [1]. Prevalence of MS is higher in females than in males in most countries in the Asia–Pacific region [2], attributed to various cultural factors including stress and low socio-economic status [3]. Besides gender, the etiology of MS is likely multifactorial and involves the interplay among lifestyle behavior, ageing, obesogenic environments, and genetic susceptibility [4], with the latter having ignited the interest for numerous genetic studies in Asia to gain insight into the genetic basis of MS and its component traits [5,6,7,8,9,10,11]. Although lifestyle is a well-known determinant for the development of MS, genetic variants, especially when they involve functional polymorphisms in the promoter regions of the genes, may alter the function and the expression of genes that are associated with energy intake and energy expenditure [12]. 

To the best of our knowledge, the genes involved with the pathophysiology of MS have not been fully elucidated. On the other hand, genome-wide studies have identified that insulin growth factors 1 (IGF1) and interleukin 6 (IL6) gene polymorphisms are responsible for metabolic traits. Pro-inflammatory cytokines play major roles in modulating lipid metabolism, adipocyte function, and insulin sensitivity [13] in addition to their anti-inflammatory and regenerative features [7]. Previous studies showed that IL6 gene-587 G/A polymorphism increased the risk of T2DM in G-allele carriers close to 1.3-fold compared to those with the C allele [8]. The single nucleotide polymorphisms (SNPs) rs1800797 (–597 G/A), rs1800796 (–572 G/C), and rs1800795 (–174 G/C) that are located in the promoter region of IL6, on the other hand, were associated with obesity and metabolic traits in different ethnic groups [12]. Additionally, rs1800796 polymorphism was associated with high insulinogenic index in Danes [14], hyperglycemia in Mexicans [15], and hypertension and obesity in Asians [16,17]. 

IGF1 plays major roles in main pathways in the progression of metabolic traits, such as progression of T2DM complications [18] and the development of cardiovascular disease [19]. It is an important growth factor linked with various biological systems, especially cell proliferation, differentiation, survival, and maturation [20]. The SNP rs35767 is known to contribute to the development of diabetes in various populations. A study showed that the TT genotype of rs35767 increases 1.92-fold and 1.77-fold of risk for T2DM and diabetic retinopathy in the Chinese Han population, respectively [21]. Besides, the CC genotype of rs7136446 has a higher body mass index and body fat percentage than TT and CT genotypes [22] but evidence is lacking on other metabolic traits [23]. Hence, the roles of IGF1 gene polymorphisms and disease susceptibility are evident and deserve further study. Table 1 summarizes the polymorphisms and their possible health consequences.

Postmenopausal women are more likely to develop metabolic traits, such as obesity and diabetes compared to men. Hormonal changes and age-related changes in body composition especially the increase in total body fat and the decrease in lean body mass may cause disturbances in glucose metabolism and insulin sensitivity [24]. However, inconclusive findings have been reported whereby the age of menopause was not related to the occurrence of diabetes among Chinese postmenopausal women [25] in China and Italy [26], probably attributed to the methodological issue [25,27]. 

In general, dietary acid load (DAL) can be categorized into acid load diet and base load diet. Acid load diet refers to a diet high in animal sources but low in vegetable and fruit. It generates anions (acidic ions such as phosphate (PO_4_^−^), sulfate (SO_4_^−^), chloride (Cl^−^), and organic acids) in the body, causing metabolic acidosis and increased urinary calcium excretion, which may have adverse effect on bone health [28]. Several studies have demonstrated that imbalance in the intake of a high acid load diet and a low base load diet might cause excess endogenous acid production which in turn leads to metabolic acidosis [29] and increases the risk of metabolic traits including obesity [30], hypertension [31], and diabetes mellitus [32], with inconsistencies reported in others [33,34]. These inconsistencies could be attributed to differences in the sample sizes, genetic heterogeneity, and dietary assessments. To date, this was the first study that has attempted to examine the interaction between the DAL with the genetic polymorphism and the development of metabolic traits among postmenopausal Chinese women. We aimed to explore the effects of DAL, IGF1, and IL6 gene polymorphisms and their potential diet–gene interactions on systolic blood pressure (SBP), diastolic blood pressure (DBP), fasting blood glucose (FBG), waist circumference, and lipid profile. 

## 2. Materials and Methods

### 2.1. Study Design and Participants

This was an analytical, cross-sectional study conducted on community-dwelling postmenopausal women in Kuala Lumpur and Selangor, Malaysia. The methodology of the study has been published elsewhere [35,36]. Briefly, a total of 211 eligible Malaysians, at least 5 years postmenopausal and with absence of severe disease were recruited based on a two-stage sampling technique. Ethical approval was obtained from the Ethics Committee for Research Involving Human Subjects (project reference number: FPSK (FR16) P019) with all respondents provided written informed consent prior to study enrollment.

### 2.2. Sociodemographic and Physical Activity

Sociodemographic information was ascertained using a pre-tested structured questionnaire as discussed elsewhere [35,36]. The physical activity level of respondents was evaluated using the Global Physical Activity Questionnaire (GPAQ) [37], with the level of physical activity was classified according to Kyu et al. [38].

### 2.3. Physical Measurement

The height of respondents was measured using a portable SECA stadiometer while the weight was measured at respondents’ fasting state using a TANITA digital weighing scale. The body mass index of respondents was computed as the ratio of weight (kg) to height in meter squared (m^2^), and the World Health Organization 2000 [39] classification was used to classify body mass index (BMI). The waist circumference (WC) of respondents was measured using Lufkin anthropometric measuring tape while blood pressure was measured using a digital automatic BP monitor (OMRON HEM-907, Omron Healthcare, Kyoto, Japan).

### 2.4. Blood Collection and Biochemical Measurement

Fasting blood samples were collected from antecubital veins in Ethylenediaminetetraacetic acid (EDTA) (Becton Dickinson, NJ, USA) and plain tubes by certified phlebotomists. The tubes were immediately placed in an icebox and transferred to an analytical laboratory at which the blood samples were separated into plasma (glucose, vitamin D, and DNA analysis) or serum (lipid profile analysis) and stored at −20 °C until subsequent analyses. Fasting plasma glucose was determined by the hexokinase method using the Olympus AU analyzer (Beckman-Coulter, Inc., Fullerton, CA, USA) while fasting serum lipid profiles (total cholesterol, triglyceride, high-density lipoprotein cholesterol (HDL-C), and low-density lipoprotein cholesterol (LDL-C)) were determined using commercially available kits on a Hitachi 704 Analyzer, which is serviced by Roche Diagnostics. Total cholesterol and triglyceride were analyzed according to cholesterol oxidase/peroxidase and the glycerol phosphate oxidase/peroxidase method, respectively. On the other hand, HDL-C was measured by direct HDL method while LDL-C was estimated indirectly using the Friedewald formula. On the other hand, serum levels of 25(OH) vitamin D was determined by using the Siemens ADVIA Centaur Vitamin D Total assay (Siemens, Tarrytown, NY, USA), with the analytical measuring range between 4.2 and 150 ng/mL (10.5 to 375 nmol/L).

### 2.5. Estimation of Dietary Acid Load

The dietary intakes of respondents were assessed using a validated semi-quantitative food frequency questionnaire (sFFQ) adapted from the Malaysian Adult Nutrition Survey 2014 [40]. The sFFQ covers 165 food items frequently consumed among Malaysians, along with their standard portion sizes. After receiving detailed instructions from researchers, respondents indicated the typical frequency of consumption of foods and average amount (in household measures, e.g., cup, bowl, spoons), to allow the estimation of food intake over the past month [41]. Portion sizes were then converted to grams, based on the published household measurement. The validity and reliability of this questionnaire among Malaysian have been assessed previously [40]. Nutrient data (protein, phosphorus, potassium, magnesium, and calcium) were then analyzed using Nutritionist Pro™ Diet Analysis (Version 3.2, 2007, Axxya Systems, Stafford, TX, USA) software, with the Nutrient Composition of Malaysia Foods [42] and Singapore Food Composition Database [43] as the primary databases. The dietary acid load of respondents was estimated according to potential renal acid load (PRAL) equation as below [44]:PRAL (mEq/d) = 0.49 protein (g/d) + 0.037 phosphorus (mg/d) − 0.021 potassium (mg/d) − 0.026 magnesium (mg/d) − 0.013 calcium (mg/d)

### 2.6. SNP Selection, Genotyping, and Quality Control Analysis

Candidate genes and SNPs were chosen from previously published literature which showed associations with metabolic traits [12,15,16,22,45,46,47,48]. IGF1 (rs35767 and rs7136446) and IL6 gene (rs1800796) polymorphisms were selected in this study. SNP sequences were referred from the website (https://www.ncbi.nlm.nih.gov/snp/). Genomic DNA was extracted from the whole blood samples (EDTA tube) using a commercially available DNA extraction kit (QIAamp DNA Blood Mini Kit Qiagen, Hilden, Germany) according to the standard protocol. The extracted DNA concentration was quantified using a spectrophotometer (Nanodrop, Thermo Scientific, Roskilde, Denmark) and the qualities of the extracted DNA were assessed using 0.8% agarose gel electrophoresis. Samples showing good quantity and quality of DNA were further analyzed using Agena^®^ MassARRAY. After the SNP detection process, Typer Analyzer was used to analyze the output data from Agena^®^ Massaray.

### 2.7. Statistical Analyses

Statistical analyses were performed using IBM SPSS 22 (SPPS Inc., Chicago, IL, USA) with the level of significance set at *p* < 0.05. The Hardy-Weinberg equilibrium (HWE) test for genotypic distribution was examined using the Hardy-Weinberg equilibrium exact test. Due to less than 5% of the respondents having TT (rs35767), CC (rs7136446), or CC (rs1800796) genotypes, they were combined with the heterozygous groups (rs35767: TT + CT; rs7136446: CC + CT; and rs1800796: CC + CG) for further analysis. Prior to analysis, data quality was analyzed via SPSS to remove outliers, handle missing values, and test normality.

Descriptive data are presented as mean ± SD and range, or percentage. Metabolic trait cut off points were followed by harmonized criteria [48] as this is the most up-to-date and is recommended by the Joint Interim Statement (JIS) committees and suitable for the Asian population [48,49,50]. Next, comparisons of respondents’ characteristics between the two groups’ genotypes of three SNPs were completed using independent Student’s *t*-test.

Furthermore, 18 models of three-step hierarchical multiple linear regression analysis were employed to test the contribution of variables as well as to determine the direct and interaction effects of DAL and genetic polymorphisms with each metabolic trait (SBP, DBP, WC, FBG, TG, and HDL and cholesterol ratios). Step one was used to determine the association between adjusted variables and metabolic traits while step two assessed the association between the DAL and gene polymorphism (rs35767, rs7136446, and rs1800796) on the dependent variable (SBP, DBP, WC, FBG, TG, and HDL and cholesterol ratios). These were followed by step three which aimed to add interaction term (DAL*gene polymorphism) and to determine the interaction effect of DAL with the gene polymorphism for each metabolic trait.

## 3. Results

### 3.1. General Characteristics

Table 2 presents the general characteristics of the respondents. The mean age of the respondents was 66.7 ± 6.6 years old while the average years of menopause were more than 15 years. Majority of the respondents attained lower secondary education with the average years of education of 8 ± 4.6 years. Approximately 99% were insufficiently active or had low activity while less than 2% were moderately active. According to the harmonized metabolic syndrome criteria [46,48], approximately half of the respondents had abdominal obesity, more than two-third had elevated SBP, and near to one-fourth had elevated DBP while more than half of the respondents had high blood glucose. Mean PRAL score was 13.8 ± 19.1 mEq/day. Among the respondents, the genotype distributions for the two IGF1 polymorphisms were TT + CT: 52.1% and CC: 47.9% (rs35767); CT + CC: 32.7% and TT: 67.3% (rs7136446). IL6 gene rs1800796 polymorphism was CC + CG: 44.1% and GG: 55.9%.

### 3.2. Demographic, Clinical, and Biochemical Characteristics According to SNPs rs35767, rs7136446, and rs1800796

Table 3 outlines the demographic, clinical and biochemical characteristics of the postmenopausal women according to the genotypes of the SNPs rs35767, rs7136446, and rs1800796. There were no significant differences in age, duration of menopause, duration of education, or BMI across the SNPs. Similarly, metabolic traits and lipid profiles (WC, SBP, DBP, blood glucose, total cholesterol, HDL, LDL, and triglycerides) were comparable between the major allele for each genotype.

Data are presented as means ± S.D., performed by independent *t*-test.

### 3.3. Contribution of Variables on Metabolic Traits

Table 4 depicts the three-step hierarchical regression analysis, highlighting the contribution of the adjusted variables of each metabolic trait. In the SBP model, age, BMI, and total cholesterol were positively associated with SBP. Thus, the higher the BMI and the total cholesterol, and the more advanced the age, the higher the SBP will be. Similarly, there were positive associations of BMI and total cholesterol with diastolic blood pressure. On the other hand, FBG, SBP, and TG were positively associated with WC while years of education were negatively associated with WC. As for fasting blood glucose, age and BMI were positively associated, while total cholesterol was negatively associated with FBG, indicating that the lower the total cholesterol, the higher the FBG. Meanwhile, HDL and cholesterol ratio and WC were positively associated with TG, while serum 25(OH) vitamin D was negatively associated with TG. Finally, ratio of HDL to cholesterol was positively associated with TG level but was negatively associated with age. Overall, these results indicate that most of the adjusted variables contributed significantly to the metabolic traits.

### 3.4. Direct Effects of DAL and Genetic Polymorphisms on Metabolic Traits

Table 3 depicts the hierarchical linear regression analyses of gene−DAL interaction effects on the metabolic traits of the respondents. There were no significant main effects of DAL and SNPs (rs35767, rs7136446, and rs1800796) on SBP, DBP, WC, TG, or HDL and cholesterol ratio. In contrast, PRAL was positively associated with FBG.

### 3.5. Gene-Diet Interaction for Metabolic Traits

There were no significant interaction effects between PRAL and IGF1 (rs35767 and rs7136446) SNPs on SBP, DBP, FBG, TG, or HDL and cholesterol ratio (Table 3). Instead, the interaction between PRAL and IL6 gene polymorphism was associated significantly with SBP (β = 0.19, *p* = 0.041). As shown in Figure 1, lower SBP was observed as corresponding to lower PRAL among the respondents with CC genotypes. On the other hand, among the G allele carriers (CG + GG), we observed that the higher the PRAL, the lower the SBP.

## 4. Discussion

There was a significant interaction effect between DAL and SNP rs1800796 with SBP trait. The present study showed that DAL may strengthen the effect of SNP rs1800796 with SBP, with the effect of GG + CG genotypes being more pronounced among those who consumed high DAL, to have higher SBP. As this is the first study investigating the interaction effect between DAL and SNP rs1800796 on SBP, the potential mechanism by which this occurs is unknown. It is hypothesized that SNP rs1800796 with GG + CG-genotype carriers may have a higher risk of hypertension if the high acidic diet is habitually consumed. Nevertheless, more work is needed before a conclusion can be drawn.

Nevertheless, contrary to expectation, this study did not find significant direct effect between DAL and SBP. This result was in contrast to the existing evidence [51,52,53]. Dehghan et al. (2019) reported that higher DAL (assessed by PRAL) was associated with 0.97 mm Hg increased in SBP [51] while Chen et al. (2019) demonstrated that higher PRAL leads to a 14% increased risk of hypertension [53]. A possible explanation for the discrepancy may be attributed to the fact that the current study was confined in its cross-sectional study design and was not able to assess the long term effect of DAL on SBP. In addition, the inconsistent result may have been due to the different genetic makeup of different ethnicities. The reason for conducting this study was to discover the different phenotypic (SBP) responses to a specific diet (DAL) depending on the genotype (IL6 and IGF1) of the postmenopausal Chinese women.

The finding of the current study did not support the association between SNP rs1800796 and hypertension as reported earlier [16]. A meta-analysis of case-control studies found that the SNP rs1800796 with G allele carriers have a higher risk of developing hypertension than C allele carriers [16]. We do not have an exact explanation for this, but it is postulated that differences in the recruited population may have contributed to the variation. 

Despite the gene interactions of DAL with DBP being statistically not significant, we do see the potential clinical significance. Globally, increased blood pressure has been reported as the top behavioral and physiological risk factor (13%) for attributable deaths [54]. Elevation of DBP (isolated diastolic hypertension or together with elevation of systolic blood pressure) is closely related to end-organ damage [55], and adequate treatment including lifestyle modifications are deemed important. To the best of our knowledge, most recommendations for lifestyle modifications have focused on reducing salt intake, weight loss, and moderation of alcohol consumption. Other dietary interventions, particularly modifying whole dietary patterns, such as the Dietary Approaches to Stop Hypertension (DASH) diet and the Mediterranean diet were effective in reducing blood pressure and thereby control hypertension. Nevertheless, the effectiveness of the DASH diet was at a smaller magnitude among normotensive individuals [56,57]. On the other hand, despite the DASH diet being a well-recognized dietary approach to reducing blood pressure, prevalence of low compliance on the DASH diet have been widely reported [58,59,60]. Moreover, decrease in compliance with time after intervention is a concern [59,61,62,63], indicating it is challenging to maintain adherence to specific dietary advice. Identification of other dietary approaches therefore merit further studies.

This study confirmed the positive association between DAL and FBG among postmenopausal women which was consistent with previous research [64,65,66,67]. The mechanism of the association between DAL and risk of type 2 diabetes has not yet been completely clarified, with several potential mechanisms being proposed. Firstly, high DAL may cause metabolic acidosis. The lower (acidic) blood pH may reduce the uptake of glucose by muscle, disrupt the insulin binding to its receptor, and further inhibit the insulin signaling pathway [68]. Secondly, high DAL may increase cortisol secretion and may consequently lead to visceral obesity and insulin resistance [69]. Thirdly, acid–base homeostasis may influence calcium and magnesium absorption in the kidney [70] and consequences of serum calcium and magnesium are related to insulin sensitivity [71,72]. Finally, yet importantly, metabolic acidosis may reduce the secretion of IGF-I [73] and inhibit insulin sensitivity [74]. On the other hand, this finding was incongruent with a longitudinal study involving 911 older non-diabetic community-dwelling Swedish men [34]. The inconsistencies in findings may be attributed to the different ethnicity, geographical distribution, age, and gender of the respondents as well as the cut-off used to determine the risk of diabetes. Further investigation of these aspects is therefore warranted.

The current findings were unable to show evidence of the association between IGF1 and IL6 SNPs (rs35767, rs7136446, and rs1800796) with FBG. In contrast to earlier findings, a meta-analysis reported a significant association between the SNP rs1800796 G allele and increased risk of T2DM among Asians [75]. Similar to SNP rs35767, previous research supports the association between SNP rs35767 and insulin resistance with C allele that may have a protective effect in T2DM [45,48]. Additionally, SNP rs7136446 is not commonly reported among the Asian population. Proposed mechanisms of the relationship between IGF1 and the risk of diabetes include IGF1 has the homology structure with insulin and is similar to insulin function that regulates hormones for insulin resistance and subsequently regulates glucose homeostasis [76]. The varying results between previous research and the current study may be due to the demographic differences discussed above. However, the major strength of the current study is the strict recruitment criteria applied. Respondents were of the same ethnicity, at least 5 years postmenopausal and without severe diseases.

This study did not support the interaction effect between DAL with gene IL6 and IGF1 on FBG, regardless of whether potential control variables were adjusted. As this is the first interaction study investigating the DAL and the SNPs on FBG, the mechanism of such an association is still unclear. The insignificant interaction indicated that the modification between DAL and SNPs is less important than other factors in determining FBG. Further investigation on other dietary patterns (such as the western diet, high-fat diet, or Mediterranean diet) with gene IL6 (rs1800796) and IGF1 (rs35767 and rs7136446) polymorphisms are merited.

The present study also investigated the potential direct effect between DAL and metabolic traits, SNPs with metabolic traits, and potential interactions between DAL and genetic polymorphisms in relation to metabolic traits. There were no significant interactions between DAL and IL6 and IGF1 SNPs on FBG, obesity or lipid traits. In addition, there were no significant direct effects between DAL on WC or lipid traits. Recent studies have shown that the findings of the association between DAL and obesity traits were contradictory. While there were positive associations between DAL and obesity traits in several studies [31,77,78,79,80], such associations were lacking in others [81]. On the other hand, studies of the associations between DAL and lipid traits are scarce. The current findings failed to confirm the association between DAL and waist circumference or lipid traits. Other unhealthy dietary patterns (such as high animal fat/calorie food and snack/fast food) or lifestyle (inactive lifestyle) are more prominent contributors to obesity and lipid traits (76–79).

As a pro-inflammatory cytokine, IL6 plays an important role in disease development. Previous research into the association between SNP rs1800796 and waist circumference or risk of obesity has been inconsistent [12,82,83,84,85]. The current study was unable to draw a definite conclusion on the associations of SNP rs1800796 with waist circumference and lipid traits among the subjects. The discrepancies in findings may be due to different ethnic populations, as variations in SNPs exist according to ethnicities and more studies are recommended.

We were not able to confirm the associations of SNPs rs35767 and rs7136446 with waist circumference and lipid traits. Nevertheless, a significant association between SNP rs35767 and total fat was observed earlier, whereby older white women with CC genotype of SNP rs35767 have 3% more trunk fat and 2% more total fat than those with C/T [47]. On the other hand, older black women with CC genotype have 3% less total lean mass and 3% less muscle mass than their TT genotype. The overall finding demonstrated that older white women and older black women with the CC genotype of SNP rs35767 have lower muscle function [47]. However, the above study was limited to the Caucasian population, with such study lacking for the non-Caucasian population. An earlier study showed that an individual with CC genotype of SNP rs7136446 had higher BMI and body fat values than TT and CT genotypes. However, such a result was not significant after being adjusted for covariates [22]. The dearth of information signifies more work is needed.

Some limitations of this study should be noted. Firstly, serum IGF1 and IL6 levels were not measured. We acknowledge that the elevation of IGF1 and IL6 levels might be able to explain better the gene–diet interaction with metabolic traits. Secondly, it was a cross-sectional study and hence unable to determine the cause and the effect of relationships between variables. Future studies should include the investigation of long-term changes of DAL on FBG and DAL/IL6 level on SBP. Additionally, we only studied three genetic polymorphisms and therefore other nucleotide changes occurring on these genes which may result in altered expression or functionality, were not investigated. This may have contributed to the small variance in explaining the model. Nonetheless, IL6 rs1800796 polymorphism and IGF1 rs35767 and rs7136446 polymorphisms were chosen as these are the SNPs common among the Asian population and had been proven to associate with metabolic traits [16,17,21,22,23]. On the other hand, adjusting potential confounding factors in the analysis to minimize the effects contributed by confounding factors is seen as one of the study’s strengths. The fact that a moderate sample size and a low ratio of some SNP percentages may have attributed to the negative correlations between metabolism traits and target genes should be considered in future studies.

## 5. Conclusions

To the best of our knowledge, this is the first study reporting the gene–diet interaction between DAL with IGF1 and IL6 gene polymorphisms on obesity, hypertension, blood glucose, and lipid traits. This study showed that higher consumption of an acid diet might increase blood glucose. In addition, the significant gene–diet interaction between DAL and IL6 on SBP, coupled with the potential clinical significance on DBP, may provide useful information in the planning for personalized nutrition.

## Figures and Tables

**Figure 1 nutrients-13-02161-f001:**
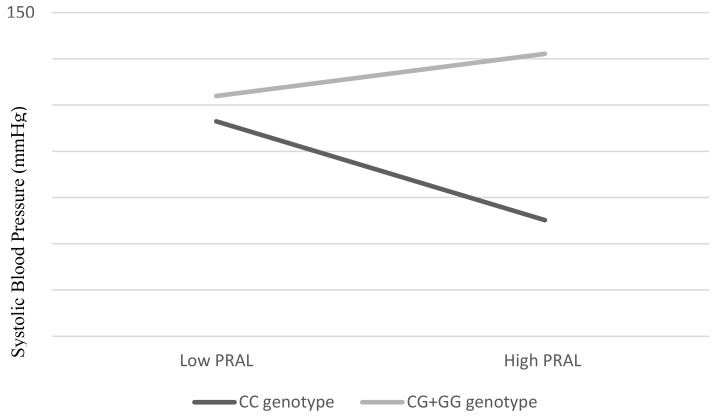
Interaction effect between PRAL (mEq/d) and SNP rs1800796 on SBP. An overall *p* value was obtained by hierarchical multiple linear regression. Estimates were adjusted for age (continuous), BMI (continuous), total cholesterol (continuous), and FBG (continuous).

**Table 1 nutrients-13-02161-t001:** Possible health consequences of different polymorphisms.

SNPs	Possible Effects on
IL6 gene -587 G/A polymorphism [8]	Diabetes
IL6 gene rs1800797 (–597 G/A), rs1800796 (–572 G/C) and rs1800795 (–174 G/C) [12]	Obesity
IL6 gene rs1800796 [14,15,16,17]	High insulinogenic index, hyperglycemia, hypertension, and obesity
IGF1 gene rs35767 [21]	Diabetes
IGF1 gene rs7136446 [22]	Body mass index and body fat percentage

SNPs: single nucleotide polymorphisms.

**Table 2 nutrients-13-02161-t002:** Demographics, anthropometrics, lifestyles, metabolic traits, and genetic analyses of postmenopausal women (*n* = 211).

	Mean ± SD or %	Range (Min–Max)
**Social demographics**		
Age (years)	66.7 ± 6.6	51–85
Postmenopausal (years)	16.1 ± 7.8	5–43
Education level (years)	8 ± 4.6	0–18
**Lifestyles**		
Physical activity (MET min/week)	1006.3 ± 875.3	0–4680
Insufficiently active (<600 MET min/week)	37.4	
Low active (600–3999 MET min/week)	61.2	
Moderately active (4000–7999 MET min/week)	1.4	
Highly active (>8000 MET min/week)	0	
**Metabolic traits**		
BMI (kg/m^2^)	24.3 ± 3.8	16.2–37.5
BMI/Obesity		
No (BMI < 30 kg/m^2^)	92.9	
Yes (BMI ≥ 30 kg/m^2^)	7.1	
WC (cm)	80.4 ± 9.3	58.4–116.7
Normal (WC < 80 cm)	50.7	
Abdominal obesity (WC ≥ 80 cm)	49.3	
SBP (mmHg)	140 ± 20.2	92–214
Normal (SBP < 130 mmHg)	31.8	
Hypertension (SBP ≥ 130 mmHg)	68.2	
DBP (mmHg)	77 ± 10.2	51–103
Normal (DBP < 85 mmHg)	76.8	
Hypertension (DBP ≥ 85 mmHg)	23.2	
Fasting blood glucose (mmol/L)	5.9 ± 0.8	4.4–9.2
Normal (FBG < 5.6 mmol/L)	40.8	
Hyperglycemia (FBG ≥ 5.6 mmol/L)	59.2	
Total-C (mg/dL)	5.8 ± 1.1	2.7–9.4
HDL-C (mg/dL)	1.6 ± 0.4	0.8–3
Normal (≥1.3 mmol/L)	89.1	
Dyslipidemia (<1.3 mmol/L)	10.9	
LDL-C (mg/dL)	3.5 ± 1	0.9–7.1
TG (mg/dL)	1.3 ± 0.6	0.4–3.8
Normal (<1.7 mmol/L)	76.8	
Dyslipidemia (≥1.7 mmol/L)	23.2	
**Dietary intake**		
PRAL (mEq/day)	13.8 ± 19.1	−49.5–85.3
Total energy intake (kcal)	1481 ± 523.6	505.8–3580.1
**Genetic analysis**		
IGF1 rs35767 polymorphism (genotype) (%)		
TT	9.4	
CT	42.7	
CC	47.9	
IGF1 rs7136446 polymorphism (genotype) (%)		
TT	67.3	
CT	28.0	
CC	4.7	
IL6 rs1800796 polymorphism (genotype) (%)		
CC	55.9	
CG	41.3	
GG	2.8	

MET: metabolic equivalent; BMI: body mass index; WC: waist circumference; SBP: systolic blood pressure, DBP: diastolic blood pressure, Total-C: total cholesterol, HDL-C: high-density lipoprotein cholesterol, LDL-C: low-density lipoprotein cholesterol, TG: triglyceride.

**Table 3 nutrients-13-02161-t003:** Demographic and metabolic traits according to SNPs rs35767, rs7136446, and rs1800796.

	IGF1 rs35767 Polymorphism		IGF1 rs7136446 Polymorphism		IL6 rs1800796 Polymorphism	
	CC genotypes (*n* = 101)	CT + TT genotypes (*n* = 110)	TT genotypes (*n* = 142)	CT + CC genotypes (*n* = 69)	CC genotypes (*n* = 118)	CG + GG genotypes (*n* = 93)
Age (years)	66.8 ± 6.9	66.6 ± 6.4	67.1 ± 6.8	66 ± 6.1	67.4 ± 7.2	65.9 ± 5.6
Duration of menopause (years)	16.6± 8.4	15.7 ± 7.2	16.3 ± 7.4	15.9 ± 8.6	16.8 ± 8.4	15.3 ± 6.9
Duration of education (years)	7.5 ± 4.4	8.4 ± 4.8	8.2 ± 4.5	7.5 ± 4.8	7.4 ± 4.5	8.6 ± 4.6
BMI (kg/m^2^)	24.3 ± 3.7	24.4 ± 3.99	24.5 ± 3.7	24 ± 3.9	24.5 ± 3.9	24.1 ± 3.7
Waist circumference (cm)	80.2 ± 88	80.6 ± 9.7	80.3 ± 9.1	80.6 ± 9.6	80.9 ± 9.7	79.7 ± 8.6
SBP (mmHg)	138.6 ± 21	141.3 ± 19.5	140.5 ± 18.8	138.9 ± 23	140.9 ± 20.8	138.8 ± 19.4
DBP (mmHg)	77.1 ± 10.2	76.9 ± 10.3	76.9 ± 10	77.2 ± 10.8	77.3 ± 10.2	76.7 ± 10.3
Fasting blood glucose (mmol/L)	5.99 ± 0.8	5.9 ± 0.9	5.9 ± 0.9	5.8 ± 0.8	5.9 ± 0.8	5.9 ± 0.9
Total-C (mg/dL)	5.8 ± 1	5.8 ± 1.2	5.8 ± 1.1	5.7 ± 1.1	5.8 ± 1.2	5.8 ± 1
HDL-C (mg/dL)	1.7 ± 0.4	1.6 ± 0.3	1.6 ± 0.3	1.7 ± 0.4	1.6 ± 0.4	1.6 ± 0.4
LDL-C (mg/dL)	3.5 ± 0.9	1.6 ± 1.1	3.6 ± 1	3.5 ± 1	3.6 ± 1.1	3.5 ± 1
TG (mg/dL)	1.4 ± 0.6	1.3 ± 0.6	1.4 ± 0.6	1.3 ± 0.6	1.3 ± 0.6	1.4 ± 0.7

Data are presented as means ± S.D., performed by independent *t*-test

**Table 4 nutrients-13-02161-t004:** Hierarchical linear regression analyses on gene−DAL interaction effects on metabolic traits.

Variable	Systolic Blood Pressure		Diastolic Blood Pressure		Waist Circumference		Fasting Blood Glucose		Triglyceride		High-Density Lipoprotein and Cholesterol RATIO	
	B Value ± SE	F Change	B Value ± SE	F Change	B Value ± SE	F Change	B Value ± SE	F Change	B Value ± SE	F Change	B Value ± SE	F Change
**Step 1 (adjusted variables)**		**22.291 ****		**10.162 ****		**17.575 ****		**8.443 ****		**18.005 ****		**17.250 ****
Age	**0.371 ± 0.184 ****		0.012 ± 0.102				**0.185 ± 0.001 ***				**−0.181 ± 0.013 ***	
BMI	**0.388 ± 0.301 ****		**0.392 ± 0.167 ****				**0.209 ± 0.001 ***					
Total cholesterol	**0.169 ± 1.100 ***		**0.154 ± 0.609 ***				**−0.191 ± 0.004 ***					
FBG	0.019 ± 17.215		−0.050 ± 9.532		**0.220 ± 7.793 ***				0.080 ± 0.508			
SBP					**0.226 ± 0.029 ****		0.023 ± 0.000		0.117 ± 0.002			
Year(s) of education					**−0.192 ± 0.123 ***						0.100 ± 0.019	
TG					**0.229 ± 0.961 ****						**0.411 ± 0.134 ****	
HDL and cholesterol ratio									**0.397 ± 0.027 ****			
WC									**0.185 ± 0.004 ***			
Serum 25(OH) vitamin D									**−0.176 ± 0.002 ***			
Physical activity											0.107 ± 0.000	
**PRAL and IGF1 rs35767 Polymorphism Model**												
**Step 2 (DAL and gene main effects)**		1.150		0.010		0.733		2.730		0.071		1.112
PRAL (mEq/day)	−0.047 ± 0.063		0.008 ± 0.035		−0.062 ± 0.030		**0.149 ± 0.000 ***		−0.011 ± 0.002		−0.045 ± 0.004	
IGF1 rs35767 polymorphism (0 = CC, 1 = TT + TC)	0.077 ± 2.348		−0.006 ± 1.307		0.045 ± 1.125		−0.027 ± 0.009		−0.019 ± 0.072		−0.077 ± 0.163	
**Step 3 (DAL*gene interactions)**		0.107		0.083		0.005		0.768		1.443		0.047
PRAL* SNP rs35767 (TT + TC)	0.033 ± 0.126		−0.032 ± 0.070		0.008 ± 0.060		−0.097 ± 0.001		0.122 ± 0.004		0.023 ± 0.009	
**PRAL and IGF1 rs7136446 Polymorphism Model**												
**Step 2 (DAL and gene main effects)**		0.395		0.296		0.521		2.797		0.451		0.511
PRAL (mEq/day)	−0.043 ± 0.063		0.009 ± 0.035		−0.058 ± 0.030		**0.149 ± 0.000 ***		−0.014 ± 0.002		−0.047 ± 0.004	
IGF1 rs7136446 polymorphism (0 = TT, 1 = CC + CT)	0.029 ± 2.528		0.049 ± 1.400		0.021 ± 1.194		0.036 ± 0.010		−0.055 ± 0.077		0.039 ± 0.176	
**Step 3 (DAL*gene interactions)**		3.814		2.082		0.525		0.612		0.217		0.048
PRAL* SNP rs7136446 (CC + CT)	−0.156 ± 0.144		−0.127 ± 0.080		−0.061 ± 0.069		0.070 ± 0.001		0.038 ± 0.004		0.018 ± 0.010	
**PRAL and IL6 rs1800796 Polymorphism Model**												
**Step 2 (DAL and gene main effects)**		0.389		0.007		0.657		2.649		1.567		1.880
PRAL (mEq/day)	−0.045 ± 0.063		0.007 ± 0.035		−0.057 ± 0.030		**0.147 ± 0.000 ***		−0.016 ± 0.002		−0.041 ± 0.004	
IL6 rs1800796 polymorphism (0 = CC, 1 = GG + CG)	0.029 ± 2.398		0.003 ± 1.331		−0.038 ± 1.136		0.010 ± 0.010		0.103 ± 0.071		−0.108 ± 0.165	
**Step 3 (DAL*gene interactions)**		**4.222 ***		0.536		0.133		1.236		0.876		0.944
PRAL* SNP rs1800796 (GG + CG)	**0.194 ± 0.124***		0.076 ± 0.069		0.036 ± 0.060		−0.117 ± 0.001		0.090 ± 0.004		−0.096 ± 0.009	

Note: * *p* < 0.05; ***p* < 0.001. bold font: significant value.

## Data Availability

Data is contained within the article or supplementary material.

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
