# Peer review of "Dietary Acid Load and Its Interaction with IGF1 (rs35767 and rs7136446) and IL6 (rs1800796) Polymorphisms on Metabolic Traits among Postmenopausal Women"

_nutrients, 2021, doi:10.3390/nu13072161_

Round 1

Reviewer 1 Report

This study was to determine association between metabolism traits with SNPs of two genes including IGF and pro-inflammatory factor IL6 in postmenopausal women population. I deem the experimental design is valid although the size of data was  low (>200 individuals). Statistical analysis was the key for this type of studies and the author has justified it well.

There are a few issues need to be resolved:

1) In Table 1, the gene names should be included along with SNPs for easy checking. 

2) In Fig. 1, there was no label for Y- although it would be told from legend. Label the curve as much as possible as this will allow author to read easier.

Again the limit of sample size and low ratio of some SNPs percentage form a major concern of this study. The negative correlation between those metabolism traits and target genes reduces the significance of this study and can't be compared with previously published results. But the discovery of association between DAL and FBG is reasonable in this designated population and support the correlation between important metabolism traits. 

Reviewer 2 Report

Nutrients-1109037. Peer review v1. 10th of February 2021.

Line 60-66. What are “TT”, “CC” and “CT”. Are authors referring to nucleosides ? Please explicit. Besides, could the authors summarise all polymorphisms (and their consequences) into one table ?

In Table 2, no statistical differences are observed. I therefore suggest to remove the caption indicating “*p<0.05 with major allele genotype”. Furthermore, what does “performed by independent t-test” mean ? The statistical test cannot be related to calculations of mean and SD. Please edit.

In Table 2, n values are difficult to understand in the genotypes. Authors enrolled a total of n=211 subjects. Yet, the numbers of polymorphisms detected is largely over 211 (101+110+142+69+118+93). Therefore, readers would assume that different genotypes can be found in a single subject. Is that correct ? Can the authors explain further ?

Again, in Table 2, a small typo is inserted in the column SNP rs1800796 CG + GG (n = = 93). Please correct the typo.

In Table 3, authors should add a small caption explaining that bold numerals are statistically significant values.

 Figure 1 should indicate the Y axis legend within the graph (Calculated PRAL, mEq/d).

Authors might want to consider removing the sentence “due to budgetary reasons, serum IGF1 and IL6 levels were not measured”. Since the funding of the study was partly governmental (line 370-372), authors cannot mention lack of funds, as the funds are attributed to specific research proposals. Therefore, I assume authors did not anticipate to measure both IGF1 and IL-6.

In the Introduction section, authors should include a small paragraph on the meaning of “dietary acid load”, as this specific notion is difficult to understand, although it is slightly explained (via an equation, lines 141-142) in the Methods section.

Finally, authors should consider revising the English throughout the manuscript.

Reviewer 3 Report

Thank you for the manuscript. 

  1. I think it is a problem that you have no plasma levels of IGF-1 and IL-6. Is it possible to measure them and add them to the paper?
  2. What is the clinical significance of the results? The associations described are weak. 
  3. What is a "high dietary acid load". You write about the term DAL, but definition is not clear. How many grams of protein per day are "too much"?
  4. The "Results" paragraph contains many results which all seem equally important. I miss that the authors focus more clearly on the main results in "Results". 

Reviewer 4 Report

The authors herein presented a paper on putative relation among DAL, il6, and igf1 polymorphisms. Although it's an interesting research field, especially toward precision medicine and targeted therapy, the study presents serious issues. 

A. The age of the group: in my opinion, the inclusion of 80 years old women in this court may provide confusing results. Please recalculate all the panels in this light. 

B. The authors look at 3 snps, not il6 and igf1 polymorphisms. It's known that other nucleotide changes occurring on these genes may result in altered expression or functionality. So, the authors should check for other alterations or, otherwise, adjust the title and the other parts of the manuscript discussing the limited number of variations that were analyzed.

Round 2

Reviewer 3 Report

Thank you very much for your fine response. 

The article has improved, and I have no further comments. 

Reviewer 4 Report

Dear authors,

Although I don't agree with the cohort composition I understand what you stated.